# Evaluation on Structural Properties and Performances of Graphene Oxide Incorporated into Chitosan/Poly-Lactic Acid Composites: CS/PLA versus CS/PLA-GO

**DOI:** 10.3390/polym13111839

**Published:** 2021-06-02

**Authors:** Siti Noor Kamilah Mohamad, Irmawati Ramli, Luqman Chuah Abdullah, Nor Hasimah Mohamed, Md. Saiful Islam, Nor Azowa Ibrahim, Nor Shafizah Ishak

**Affiliations:** 1Department of Chemistry, Faculty of Science, Universiti Putra Malaysia (UPM), Serdang 43400, Selangor, Malaysia; kamilahmohamad@rocketmail.com (S.N.K.M.); norazowa@gmail.com (N.A.I.); 2Department of Chemical Engineering, Faculty of Engineering, Universiti Putra Malaysia (UPM), Serdang 43400, Selangor, Malaysia; chuah@upm.edu.my (L.C.A.); norshafizah@gmail.com (N.S.I.); 3Malaysia Nuclear Agency, Kajang 43000, Selangor, Malaysia; shima@nuclearmalaysia.gov.my; 4Department of Chemistry, Bangladesh Army University of Engineering & Technology, Qadirabad, Natore 6431, Bangladesh; msaiful2007@gmail.com

**Keywords:** chitosan, poly(lactic acid), graphene oxide, blending, composite materials

## Abstract

In this work, to fabricate a novel composite consisting of chitosan/poly-lactic acid doped with graphene oxide (CS/PLA-GO), composites were prepared via solution blending method to create various compositions of CS and PLA (90/10, 70/30 and 50/50CS/PLA-GO). Graphene oxide (GO) was added into a PLA solution prior to blending it with chitosan (CS). The surface morphology and structural properties of synthesized composites were characterized using FT-IR, SEM and XRD analysis. The performances of synthesized composites on thermal strength, mechanical strength, water absorption, and microbial activity were also evaluated through standard testing methods. The morphology of 70/30CS/PLA-GO became smoother with the addition of GO due to enhanced interfacial adhesion between CS, PLA and GO. The presence of GO has also improved the miscibility of CS and PLA and has superior properties compared to CS/PLA composites. Moreover, the addition of GO has boosted the thermal stability of the composite, with a significant enhancement of *T_d_* and *T_g_*. The highest *T_d_* and *T_g_* were accomplished at 389 °C and 76.88 °C, respectively, for the 70/30CS/PLA-GO composite in comparison to the CS and PLA that recorded *T_d_* at 272 °C and 325 °C and *T_g_* at 61 °C and 60 °C, respectively. In addition, as reinforcement, GO provided a significant influence on the tensile strength of composites where the tensile modulus showed remarkable improvement compared to pure CS and CS/PLA composites. Furthermore, CS/PLA-GO composites showed excellent water-barrier properties. Among other compositions, 70/30CS/PLA revealed the greatest decrement in water absorption. From the antibacterial results, it was observed that 90/10CS/PLA-GO and 70/30CS/PLA-GO showed an inhibitory effect and had wide inhibition zones which were 8.0 and 8.5 mm, respectively, against bacteria *Bacillus Subtillis* B29.

## 1. Introduction

Chitosan (CS) is a natural biopolymer and has great potential as an antimicrobial packaging material, coating or blending with a wide array of pathogenic bacteria and fungi that are inhibited on direct contact with CS [1]. Due to the chemical structure of CS comprising of one NH_2_ group and two OH groups on every glucosidic residue, it has a cationic structure [2]. The presence of the NH_2_ group in CS makes it more adaptable to a reaction than other natural polymers. CS-based composite is known as a composite that has superior physical and chemical properties such as porosity, tensile strength, photo-luminescent, high surface area and conductivity as well as mechanical properties [3]. Chitin is the second most abundant natural polymer after cellulose [4], and CS is derived from chitin (exoskeletons such as prawns, crustacean shells) via deacetylation reaction; thus, it is normal that CS has received attention from many researchers. CS is easy to obtain and is also a very compatible and effective biomaterial usable for many applications, for example, in the biomedical field [5,6], where it serves as an alternative candidate for drug delivery because it has a regenerative effect on connective tissue [7]. CS is normally associated with a moisture-resistant polymer while maintaining the overall biodegradability of the product. However, CS is generally low due to hydrophilic properties and pH sensitivity. By associating CS with other materials, it has been proven to be efficient to improve the mechanical and chemical properties of CS. Association between polymers can become blends or multilayer products, for example, in coating or laminating, but blending is the easier and fastest way to prepare multiphase materials with desirable properties.

Blending chitosan with poly-lactic acid (PLA) is an option [8]. PLA belongs to the family of aliphatic polyester commonly made from renewable resources of lactic acid such as starch via fermentation processes [9]. PLA is a thermoplastic, high-strength, non-toxic, natural, biodegradable and high-modulus polymer. It has also been used in several applications such as food packaging, water and milk bottle, barriers for sanitary products and diapers as well as in automotive applications [10,11]. PLA has hydrophobic properties due to the existence of the methyl group (−CH_3_) which is a non-polar and covalent bond. However, CS and PLA have different polarities as CS can dissolve in dilute acidic solutions because of its hydrophilic properties, whereas PLA can dissolve in organic solvents such as chloroform due to its hydrophobic properties [12].

Nonetheless, CS is immiscible with PLA and it is necessary to use a third polymer or compatibilizer to reduce the particle size of the dispersed phase and to enhance the attractive interaction between the discrete phase so that a compatible blend can be achieved [13].

There were several studies conducted on adding compatibilizer or other polymers into the CS/PLA blend, such as adding tea polyphenol (TP) into the CS/PLA blend composite [14]. Results showed that PLA/CS/TP addition of TP significantly increased the heat-sealing strength, water vapor permeability and solubility of the composite PLA/CS/TP membrane.

Nevertheless, there is no research reporting on a CS/PLA-GO composite. Thus, in this study, graphene oxide (GO) was introduced to a CS/PLA composite as a compatibilizer. Recently, graphene oxide (GO) has received a lot of attention for many applications due to its potential as the material with the utmost electrical mobility at room temperature, being very elastic and holding excellent thermal conductivity [15]. Due to their large surface area, excellent mechanical properties, high stability, good electrical conductivity and ease of functionalization, graphene, graphene oxide (GO) and functionalized GO nanosheets have been impressively considered as a great promising compatibilizer [16]. GO also has properties such as mechanical stiffness and extraordinary electronic transport properties which make GO a good material for the development of conductive platforms [17]. According to recent studies, adding graphene-based materials GO, GO modified with silane, GO modified with stearic acid and GO modified with poly(ethylene glycol) (PEG) to a PLA matrix significantly improved the mechanical efficiency, thermal properties, gas barrier and water vapor permeability of the resulting final product. In this regard, the appropriate functionalization of the GO surface becomes a critical factor in achieving compatible interfaces that facilitate good compatibilizer dispersion into the PLA matrix, resulting in improved final product properties. Since GO contains numerous hydrophilic groups, such as hydroxyl, carboxyl and epoxy groups [18], it is simple to graft macromolecules onto its surface through covalent or noncovalent functionalization. In biomedical applications, functionalizing GO with a biocompatible polymer will significantly reduce the cytotoxicity of GO to cells and animals [16]. Graphene also has the ability to interact with tissues, biomolecules and cells which adds another application of GO in various biomedical fields such as drug delivery and tissue engineering [17]. Nanographene oxide/chitosan influences the adhesion and proliferation rate of nerve cells with a 20% increase in nerve cell growth [19]. CS/GO nanocomposite membrane has improved in congo red dye absorption uptake with >90% removal efficiency [20]. The addition of GO without surface modification into polymers has also suggestively affected the thermal and mechanical properties of the polymers to some degree [21].

Herein, this research aims to study the effect of GO into a CS/PLA composite via the blending method and to study its performances in terms of thermal stability, water adsorption and antimicrobial activity of the composites (Scheme 1). The results might eventually lead to new alternatives in numerous applications such as biomedical.

## 2. Materials and Methods

### 2.1. Chemical and Reagents

Chitosan (degree of deacetylation = 75–86% low molecular weight, average *M*_w_ = 50,000–190,000), poly(L-lactide) (product number 764590, average *M*_w_ = 5000) and graphene oxide (product number 796034, 15–20 sheets, 4–10% edge-oxidized) were purchased from Sigma-Aldrich, Darmstadt, Germany. The analytical grade solvents used were chloroform and acetic acid supplied by Sigma-Aldrich, Darmstadt, Germany.

### 2.2. Synthesis of Composites

#### 2.2.1. Fabrication of CS/PLA Composites

90% *w/w* of chitosan (CS) was dispersed in 100 mL of acetic acid solution. A unit of 10% *w/w* of poly-lactic acid (PLA) was dispersed in 100 mL chloroform. The two solutions were mixed and stirred for 24 h at room temperature. The blended CS/PLA composite was cast in a Petri dish at the size of 90 × 15 mm and dried in a drying oven at 62 °C. The composite was kept dry in a desiccator for solvents to evaporate. These steps were repeated for various proportions of CS/PLA which are 70/30 and 50/50% *w/w*. The samples were denoted as 90/10CS/PLA, 70/30CS/PLA and 50/50CS/PLA, respectively.

#### 2.2.2. Fabrication of CS/PLA-GO Composites

90% *w/w* of CS was dispersed in 100 mL of acetic acid solution. A unit of 10% *w/w* of PLA was dispersed in 100 mL chloroform, and 2% *w/w* of graphene oxide (GO) was added into the PLA solution and stirred for 3 h at room temperature. The two solutions were mixed and stirred for 24 h at room temperature. The blended CS/PLA composite was cast in a Petri dish at the size of 90 × 15 mm and was left dry in a desiccator for 3 days. The procedures were repeated for various proportions of CS/PLA-GO which are 70/30 and 50/50% *w/w* while GO loading was kept constant. The samples were denoted as 90/10 CS/PLA-GO, 70/30 CS/PLA-GO and 50/50 CS/PLA-GO, respectively.

### 2.3. Composite Characterizations

Fourier transform infrared (FTIR) spectroscopy was performed using a Perkin Elmer 1725× Spectrometer made in China. The composite samples were dissolved and mixed with the KBr particles together and pressed into pellets. The spectra of the composite samples were recorded in the range of 4000–400/cm.

The surface morphology of the sample was determined using the JEOL Model JSM-6400 scanning electron microscope (SEM) made in Japan. The samples were placed on the stub and coated with gold. The images were scanned at a magnification of 500×.

The degree of structural order (crystallinity) of composite samples was examined using an X-ray diffractometer (XRD). The X-ray diffraction of the samples was observed with a Shimadzu 6000 X-ray diffractometer with nickel-filtered Cu Kα (λ = 1.542 Å) radiation. The scanning angle (2θ) of the XRD was at the range of 0–40 °C at the rate of 4 °C/min.

Thermal stability of composites was measured through Thermogravimetric Analysis (TGA) and Differential Scanning Calorimetry (DSC) techniques. TGA measurements were carried out on 2 mg of each sample at a heating rate of 10 °C/min with nitrogen atmosphere using a Thermogravimetric Analyzer a Perkin-Elmer Pyris 7 made in Japan. Thermal decomposition of each sample occurred in a programmed temperature range of 30–800 °C. The continuous weight loss and temperature were recorded and analyzed.

The DSC test was conducted using a Perkin Elmer thermal analyzer: a Perkin-Elmer Pyris 7 made in Japan. All measurements were made under a nitrogen flow (50 mL/min), keeping a constant heating rate of 10 °C/min while using an alumina crucible with a pinhole.

Mechanical strength was tested using the Instron 4400 Universal Tester, Jinan, China to measure the tensile strength at the point of breakage for each sample. Tensile tests were carried out at room temperature, according to ASTM D882 type V. A fixed crosshead of 10 mm/min was utilized in all cases and the results were taken as an average of five tests.

Following ASTM D570, spherical specimens were prepared for each different composite having a diameter of 6 mm. The specimens were dried in an oven at 60 °C for 1 h and left to cool in a desiccator. The specimens were weighed immediately upon cooling and immersed in deionized water at room temperature for 24 h. The specimens were removed, patted dry and weighed. After weighing, the specimens were re-immersed back into water and reweighed for 5 consecutive days. Five test specimens of each composite were prepared for the test and the results were averaged. Water absorption is expressed as the increase in weight percentage and was calculated according to the formula shown below in equation Equation (1):(1)$$\mathrm{Water}\;\mathrm{absorption}\;\left(\%\right)=\frac{w_{2}-{\;w}_{1}}{w_{1}}\times 100\%$$
where w_1_ is the dry weight of the specimen; w_2_ is the wet weight of the specimen.

To evaluate in vitro antibacterial activity, the samples were cold-pressed into paper discs with a diameter of 8 mm containing *Bacillus Subtillis B29*, *Pseudomonas aeruginosa ATCC 15442, Salmonella choleraesuis ATCC 10708* and *Candida tropicalis A3*. The microbe culture was standardized to 0.5 McFarland standard which was approximately 108 cells. The plates were inverted and incubated at 30–37 °C for 18–24 h, 24–48 h or until sufficient growth occurred. After incubation, each sample was examined. After completion, sliding calipers or a ruler were placed under the disc and the diameters of the discs and the inhibition zones were measured and determined in millimeters (mm).

## 3. Results

### 3.1. Characterization of CS/PLA and CS/PLA-GO Composites

#### 3.1.1. Fourier Transform Infrared Spectroscopy (FTIR)

FTIR spectra of CS, PLA and all synthesized composites are presented in Figure 1 and Appendix A. As can be seen in Figure 1a, the PLA spectrum clearly shows the characteristic absorption band located at 2997 and 2947 cm^−^^1^ due to the C–H asymmetric stretching vibration. While the peak at 1755 cm^−^^1^ was due to the vibration of −C=O bonds of the carbonyl group [22], the peak at 1454 cm^−^^1^ was due to C–H stretching in the CH_3_ [23]. In Figure 1b, the spectrum of CS clearly shows the asymmetric broadband due to the overlapping of O–H and N–H stretching bands extending from 3313 to 3278 cm^−^^1^ [24]; 2916 cm^−^^1^ for CH stretching: two middle strong bands at 1643 and 1546 cm^−^^1^ (amide I and amide II) [25], bands of CH_n_ groups at 1404 cm^−^^1^ (CH_2_ deformation) and 1369 cm^−^^1^ (–CH_3_ symmetric deformation) [26] and peaks at 1068 and 1022 cm^−^^1^ were due to C–O stretching. In Figure 1b, it can be observed that noticeable changes occurred in the spectrum of CS/PLA in comparison with the spectrum of each component. The two original bands of the chitosan component at 1643 and 1546 cm^−^^1^ for amide I and amide II shifted to lower wavenumbers for 90/10CS/PLA, 70/30CS/PLA and 50/50CS/PLA composites. An original strong peak of the PLA component at 1755 cm^−^^1^ for the ester group became significantly weaker and was markedly wider. The intensity of the stretching bands overlapped and centered near 3313 and 3278 cm^−^^1^ in chitosan for the hydroxyl and amino groups pronouncedly decreased. All these registered events indicate that there were obvious interactions among the amino, carboxyl and hydroxyl groups of the two components inside the composites. These interactions should be attributed to the hydrogen bonds possibly forming between amino (in CS) and carboxyl (in PLA) of hydroxyl (mainly in chitosan) and carboxyl groups because there was no covalent interaction between CS and PLA chains [27]. Interaction between CS and PLA was due to H bonds.

As the GO was being loaded to CS/PLA, the broadband intensity of the stretching bands overlapped and centered near 3313 and 3278 cm^−^^1^, representing −OH and −NH_2_. Nonetheless, there was a deformation in the vibration peak of (−NH−) groups. This might be due to GO functionalities with the CS groups causing a nucleophilic reaction. The 90/10 CS/PLA-GO observed a main peak at 1068 and 1369 cm^−^^1^ which corresponds to the stretching C−O−C bonds of alkoxy indicating the epoxy functionalities of GO. The intensity of the epoxides strengthened substantially upon reduction of CS loading. The peaks at 1546 and 1643 cm^−^^1^ related to the characteristics of carbonyl and carboxyl functionalities, respectively. Moreover, the addition of GO into the oxygen-containing groups improved the miscibility and dispersion of GO in the matrix via hydrogen bonding/electrostatic interaction.

#### 3.1.2. Scanning Electron Microscope (SEM)

Figure 2 shows the surface morphology of CS, PLA and all synthesized composites. As can be seen in Figure 2a, the morphology of CS samples shows homogeneous, dense and smooth surface structure whereas in Figure 2b, the PLA’s fractured surface is covered with uneven fibrils and numbers of microvoids throughout the surface. The fractured surface of pure PLA could be categorized as ductile fracture [28]. Figure 2c–e shows blended CS/PLA without GO at the composition of 90/10, 70/30 and 50/50, respectively. As can be seen from the SEM micrographs, CS/PLA formed a smooth surface composite indicating an interaction between CS and PLA except for 50/50 composites.

Meanwhile, Figure 2f–h shows CS/PLA blended with GO at the composition of 90/10, 70/30 and 50/50, respectively. The morphology shows a smooth surface compared to the blends without GO except for 50/50CS/PLA and 50/50CS/PLA-GO which show no difference. In correlation with the FT-IR spectrum, this indicates that there is better interaction between CS/PLA with the addition of GO. Thus, it can be concluded that the addition of GO into PLA has improved the miscibility between CS and PLA. This is due to the various groups in the GO that help PLA to dissolve well in hydrophilic solutions. GO holds some oxygen-functional groups such as epoxy, hydroxyl and carboxyl groups between lamellas, which can easily be exfoliated and functionalized to form homogeneous suspensions in both water and organic solvents [29].

#### 3.1.3. X-ray Diffraction (XRD)

The XRD patterns of CS, CS/PLA and CS/PLA-GO at various compositions of 90/10, 70/30 and 50/50 composites are shown in Figure 3. The XRD patterns of CS show two characteristics peaks at 11.7° and 18.28°. The crystalline peak centered at around 11.7° is attributed to the hydrated crystalline structure of CS while the crystalline peak at 18.28° is reported to be an indication of the relatively regular crystal lattice of chitosan [30]. It is known that CS always contains bound water even when extremely dried. The incorporation of bound water molecules into the crystal lattice, commonly termed hydrated crystals, generally gives rise to a more dominated polymorph which can normally be detected by a broad peak in the corresponding XRD patterns.

The XRD pattern of 90/10CS/PLA shows the crystalline peak at 11.7° and 18.3° corresponding to the peak in CS. On the other hand, XRD patterns of 70/30CS/PLA and 50/50CS/PLA peak at 16.88° and 16.96°, respectively. The crystalline peak around 16.78° was corresponding to a broad diffraction peak of PLA [31,32]. For both 70/30CS/PLA and 50/50CS/PLA composites, the peaks of CS did not appear in their XRD pattern. Diffraction peaks disappeared in composites, indicating that intercalation or exfoliation structures have been formed.

As can be seen in Figure 3, a distinct broad peak at 8.84° can be seen in the XRD pattern for 90/10CS/PLA-GO, indicating the presence of GO, and other peaks at 11.92°, 16.8° and 18.82° correspond to the CS and PLA individual peaks. Despite individual peaks of CS and PLA appearing in the 90/10CS/PLA-GO composite, there was an improvement in the crystallinity and interaction of CS/PLA due to the presence of GO. This can be seen in the reduction of peak intensity at 11.70° and 18.28°. The peak of 11.7° and 18.28° also shifted to slightly bigger diffraction of 11.92° and 18.82°, respectively. The original crystalline structure of CS could have been partially destroyed or seriously modified, and consequently, the blend composites obtained a partially miscible structure with more amorphous domains due to the interactions between CS and PLA. As can be observed for 70/30 CS/PLA-GO, only one distinct peak formed at 16.98°, while 50/50 CS/PLA-GO gave two crystalline peaks at 16.86° and 18.20°. XRD patterns for 70/30 CS/PLA-GO and 50/50CS/PLA-GO evidence that CS effectively blended with PLA with the addition of GO except for 90/10CS/PLA-GO. Apparently, if no intermolecular interaction exists between the two components, each component will form its own crystalline domains, respectively. From the result, some distinct changes in diffractograms of CS/PLA-GO composites are observed, indicating that CS, PLA and GO interacted with each other in a certain manner. As for the GO peak reported in [33], the broad peak at 26° did not show in any XRD patterns.

### 3.2. Performance of Synthesized CS/PLA and CS/PLA-GO Composites

#### 3.2.1. TGA and DSC Result Analysis

Figure 4 shows TGA thermograms for CS, PLA, CS/PLA and CS/PLA-GO at various compositions. From Figure 4, it can be seen that thermal degradation of CS and PLA takes place in two-stage weight loss and single-stage weight loss, respectively, which can be evidenced from their DTG curves in Figure 5. There are two stages of weight loss for the pure CS sample. The first degradation starts from 94 °C–175 °C in the form of a 10% loss in weight due to dehydration. The second weight loss in the range of 230 °C–371 °C corresponds to 68% of weight loss attributed to the decomposition of the CS main chain. The weight loss of CS at 600 °C was 77%. The remaining residue of CS was about 23%, mostly due to the bonding of C, N and O in the CS structure. The maximum decomposition for CS is at 272 °C [34]. Whereas for PLA, a deep decomposition was evidenced from the DTG curve around 315–476 °C, where the maximum decomposition temperature was 325 °C [34]. All residue was removed as all the materials decomposed by 535 °C. As for the thermal degradation of GO, the fast decomposition of the GO framework by oxygenated groups occurred at a temperature range of 120 to 300 °C. GO underwent a dramatic 19% weight loss from 120 °C to 300 °C, and reached 28.3% weight loss at 600 °C because of the pyrolysis of the labile oxygenic functional groups on the GO surface [35,36].

Meanwhile, Figure 5 shows the DTG thermograms of CS, PLA and CS/PLA composites and CS/PLA-GO composites. As can be seen in Table 1, the temperature at maximum degradation rate (*T_d_*) in CS/PLA-GO composites is higher compared to *T_d_* (temperature at maximum degradation rate) in CS/PLA composites. This indicates that the addition of GO into CS/PLA composites will increase the efficiency interaction between the two components [37]. This is due to several functional groups in the GO such as carboxylic acid which helps to increase the hydrophilic property of PLA as GO was blended with PLA prior to blending with chitosan. Increased PLA hydrophilicity reduces immiscibility between CS and PLA, resulting in increased contact between the two components.

Figure 6 shows the DSC thermograms of CS/PLA composites and CS/PLA-GO composites. For blend composites, if no interaction exists between the two components, the DSC thermograms will show two onset temperatures for each CS and PLA. In Figure 6, all CS/PLA composites have only one onset temperature of thermal degradation (*T_onset_*); thus, it can be deduced that there was an interaction between CS and PLA due to hydrogen bonding. All composites exhibit higher *T_onset_* (the onset temperature of thermal degradation) compared to CS components but lower than the *T_onset_* of PLA. All the data regarding their *T_onset_* and *T_d_* are summarized in Table 1. The maximum degradation temperature (*T_d_*) also increased by blending CS with PLA. The higher maximum degradation temperatures may be attributed to the increase in molecular weight due to an interaction between CS, PLA and GO. In thermal degradation of the polymer, there are three stages: depolymerization, random chain scission and side-group elimination [38]. As a result of the interaction between CS, PLA and GO, the maximum degradation temperature rises as the molecular weight rises. Furthermore, the presence of GO is distributed uniformly in the CS/PLA composites, acting as a barrier to the permeability of volatile degradation products out of the composite materials, delaying the thermal-degradation process [31]. These results indicate that the components are well blended together. This is because the molecular chains of CS and PLA can become deeply entangled with each other during solution-processing procedures, preventing them from efficiently recrystallizing after they are formed into the membrane and dried.

An important parameter frequently used to assess whether two polymers are miscible in the amorphous phase is the glass transition temperature *T_g_* of the composites. Only one glass transition temperature will show in the composites’ DSC thermograms if two polymers are completely miscible with each other and blended into a new one. If the two polymers are partially miscible, the resulting blend composites will have two *T_g_*’s related to each of the components, but the *T_g_* reading will be affected by the other one. Figure 6 shows DSC thermograms of CS, PLA, GO, CS/PLA and CS/PLA-GO. In the DSC thermograms of CS, a wide endothermic peak centered around 93 °C with a large interval is probably due to the absorbed moisture, and another exothermic peak centered at near 278 °C is probably linked to the decomposition procedure of CS which started at around 250 °C and is also basically in good agreement with the TG analysis as shown in Figure 4. *T_g_* CS is said to be not reactive enough to be detected by the DSC analysis. This could be ascribed to the structure and properties of CS. It is known that CS is a semicrystalline polymer due to its strong intramolecular hydrogen bonds on the backbone, and it has a rigid amorphous phase because of its heterocyclic units. Thus, when CS is heated within a certain range of temperature below its decomposition temperature, the variations in heat capacity corresponding to the change in specific volume near *T_g_* are probably too small to be detected by the DSC technique. Fortunately, in this study, the *T_g_* of CS was observed by direct and careful measurement of the differential scanning calorimetry (DSC), which was assumed to be not sensitive enough to detect the *T_g_* for CS. The successfully obtained *T_g_* of CS in our study is 61 °C. However, it is difficult to compare the *T_g_* of CS to the previous study because a few studies on CS reported different values of *T_g_* such as 70, 150, 161, 203 and 240 °C. On the other hand, *T_g_* for neat PLA was successfully obtained at baseline steps around 60 °C. Another peak recorded at around 159 °C can be assigned to its melting point temperature *T_m_*. All blend composites were measured using the DSC technique and the data are shown in Figure 6 and the statistical analysis of the data shown in Figure 7. The data on the *T_g_* of each component in the blended composite are collected and listed in Table 1. The increased *T_g_* for blend composites compared to CS can be seen in Table 1 except for 90/10CS/PLA. This suggests that a reduction in the mobility of the CS chains may be caused by their immobilization on the surface of PLA. The *T_g_* for 90/10CS/PLA is lower compared to CS but with the addition of GO into 90/10CS/PLA-GO, the *T_g_* increases slightly higher than CS. Herein, the addition of GO into the blends increases the glass transition temperature of the composite, which results in GO helping to immobilize chains of CS and PLA.

#### 3.2.2. Mechanical Strength Analysis

The tensile strength of the synthesized composites at various compositions is displayed in Figure 8. 90/10CS/PLA-GO recorded a tremendous result compared to 70/30CS/PLA-GO and 50/50CS/PLA-GO. The high loading of CS on PLA-GO has increased the compatibility and adhesion between CS, GO and PLA, thereby resulting in higher tensile strength for the composite. It is reported that the uniform dispersion of highly compatible filler as can be seen in the SEM images will lead to good interaction with the polymer matrix, which then leads to the increase of mechanical integrity and thermal stability of the composites [39].

#### 3.2.3. Water Absorption Analysis

The water absorption patterns of CS and manufactured composites are shown in Figure 9. It was observed that the water absorption of the composites increased and decreased inconsistently throughout the test. A different amount of water was absorbed by the composites every day. CS showed high water absorption due to high sensitivity to moisture caused by protonation of its primary and secondary amino and hydroxyl groups, ranging from 0 to 3.16%. As shown in Figure 9, water absorption of CS/PLA composites ranging from 0 to 1.34% showed a significant decrement compared to CS that was agreed by the previous study reported [40], where the CS/PLA composite showed the water sensitivity of pure CS film decreased with the incorporation of PLA. This implies that the integration of PLA with CS allows the composites to acquire better water barrier characteristics. Nevertheless, the incorporation of GO into the different composition of CS/PLA composites led to an even higher decrement in water absorption in CS/PLA-GO composites ranging from 0 to 0.43% only. The existence of GO resulted in a less-coiled polymeric structure and fewer interactions between chains, reducing the blocked water-absorbing sites.

The reduction percentage of water absorption within 5 days with respect to the CS sample was measured for all synthesized composites. The purpose reduction percentage of water absorption was tested within 5 days to identify an average reading of percentage in water reduction as well as to reduce the human error in this analysis. As can be observed in Figure 10, the highest percentage in water reduction can be observed in 70/30CS/PLA-GO. The reinforcement of GO lowers the porosity of CS/PLA-GO, therefore leading to a lower rate of open pores for water absorption. Herein, GO incorporated into CS/PLA-GO possesses good water barrier properties than CS/PLA.

#### 3.2.4. Antimicrobial Activity

The bacteria *Bacillus Subtillis B29*, *Pseudomonas aeruginosa ATCC 15442, Salmonella choleraesuis ATCC 10708* and *Candida tropicalis A3* were used in this antimicrobial activity. However, out of all the bacteria, the only positive result achieved was against *Bacillus Subtillis B29.* The images of the inhibition zones for CS and all synthesized composites at various compositions are shown in Figure 11. CS is known to naturally possess at least a mild antibacterial effect and it is shown in Figure 10. All CS/PLA composites did not have any antibacterial activity. Meanwhile, 90/10CS/PLA-GO and 70/30CS/PLA-GO composites showed inhibition zones against *Bacillus Subtilis B29*. On the contrary, 50/50CS/PLA-GO did not show any antibacterial activity: this might be because the lower amount of CS used causes a lower amount of interaction between CS and GO.

This suggests that at the optimum amount of CS/PLA composition loaded with GO, it could be an effective antibacterial agent. GO has been proven to have high antibacterial activities [41]. The images above can be supported by the growth inhibition value measured. There are zero inhibition zones measured for CS, 50/50CS/PLA-GO, 90/10CS/PLA, 70/30CS/PLA and 50/50CS/PLA. Only 90/10CS/PLA-GO and 70/30CS/PLA-GO showed an inhibition zone and were successfully measured at 8.0 mm and 8.5 mm, respectively. Thus, the present study showed the addition of GO into CS/PLA composites, paving the way towards further use in the field of biomedicals such as drug deliveries and food packaging. In the future, further antimicrobial and antioxidant studies on the composites will help to identify the applications suitable for the composites. The test was carried out by placing an 8 mm diameter paper disc containing composites onto a plate on which the microbes were growing. The microbe culture was standardized to 0.5 McFarland standards at approximately 10^8^ cells. Streptomycin standard was used for bacteria *Bacillus Subtilis B29.* The plates were inverted and incubated until sufficient growth had occurred and each plate was examined. The diameter of the zones of complete inhibition was measured, including the diameter of the disc. Zones were measured to the nearest whole millimeter, using sliding calipers or a ruler, which was held on the back of the inverted Petri plate.

## 4. Conclusions

A blend of composite films composed of CS/PLA and CS/PLA-GO with various compositions were successfully prepared using a solution-blending technique. The results successfully showed that the incorporation of GO into CS/PLA indicated some level of attractive interaction between CS, GO and PLA. SEM results revealed that the composites were well blended and yielded more homogenous composites. The presence of GO also improved the miscibility of CS and PLA and had better properties compared to CS/PLA composites. The addition of GO increased the thermal stability of the composite, increased the maximum degradation rate temperature (*T_d_)*, and caused a significant increment in the glass transition temperature (*T_g_*). The 70/30CS/PLA-GO composite recorded the highest *T_d_* and *T_g_* at 389 °C and 76.88 °C, respectively. The addition of GO also caused a significant impact on tensile strength where the tensile modulus showed improvement compared to pure CS and synthesized CS/PLA composites. Moreover, CS/PLA-GO composites showed good water barrier properties compared to pure CS and CS/PLA. Among other compositions, 70/30CS/PLA showed the highest decrement in water absorption. 90/10CS/PLA-GO and 70/30CS/PLA-GO showed good antibacterial activity and wider inhibition zone at 8.0 and 8.5 mm, respectively. Herein, it can be deduced that GO reinforced into CS/PLA composites improved the properties of CS/PLA in terms of mechanical strength, water resistance and microbial activity, which can be applied in medical products. More studies need to be conducted to investigate the biological performance of the composites, which will be carried out in our future research works.

## Data Availability

The data presented in this study are available on request from the corresponding author.

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
