# Peer review of "Evaluation on Structural Properties and Performances of Graphene Oxide Incorporated into Chitosan/Poly-Lactic Acid Composites: CS/PLA versus CS/PLA-GO"

_polymers, 2021, doi:10.3390/polym13111839_

Round 1
Reviewer 1 Report
Dear Sirs,
The article entitled “Evaluation on Structural Properties & Performances of Graphene Oxide Incorporated into Chitosan/Poly-Lactic Acid Composites: CS/PLA vs. CS/PLA-GO” has the main goal of characterizing GO-doped CS/PLA composites. Content is interesting and relatively new, but its strength insufficiently detailed. Please compare your work to others in the literature, name the ones with DOI: 10.1002/pat.5202, 10.3390/ma12132077 and 10.1016/j.ijbiomac.2017.08.048 among others and reinforce the novelty of your article.
I personally dislike abbreviations, etc. in the title. I would recommend “and” instead of “&” and “versus” instead of “vs.”. I would add the potential application in the title.
Please revise your text to correct distraction and grammar mistakes.
Abstract:
What is the architecture of the composites? Film, hydrogel? How were the solutions processed after blending? You analysed then the surface of what exactly?
Line 18: How was GO added? Was it added to PLA solution prior blending it to CS?
Line 21: methods
Line 22: enhanced
Line 26: …, in comparison to the control (…ºC)
Line 32: against which bacteria?
Description of the novelty and prospective applications are missing.
Introduction:
I would start the introduction with the problem that you would like to solve. Description, prevalence, severity, current solutions, need for new approaches.
CS’s mechanical properties are not the greatest… you need to explain this further to make sense. But if you are talking about CS-based composites, you need to detail which ones exactly, first pinpointing CS’s limitations alone.
Line 54: for what? You need 1. To describe potential applications and 2. Chose a potential application to give direction to your text. Otherwise, it is too fragmented.
Line 70: only tea polyphenol? Is it TP or PA? Both CS and PLA are widely studied biomaterials. You need to complement your explanation a little further.
Lines 84-85: I understand that you didn’t do deep into the intended application. But that is essential. You need to have a direction for your study. Please reinforce it, so that your work can gain much more strength.
Materials and Methods:
Line 147: in vitro
Line 148: above, you need to justify the choice of B. subtilis for your biological testing; 30-37ºC
Line 153: ?
Results and Discussion
Figure 1: with some arrows or lines, and text boxes, you can highlight spectral differences and relevant peaks. The same for the other plots.
Antibacterial testing is the one that incites doubts, in my opinion. Other than that, very well explained. Please improve image quality, add overviews, arrows or lines highlighting the inhibition zone for clear interpretation. I cannot see what you wrote.
Conclusions
Line 409: explain better processing methodology used. The second sentence is also a bit confusing.
It is generally well-written. The only thing missing is an indication of the novelty of your work. Otherwise, it is just another film for food packaging or medical applications.
Reviewer 2 Report
The manuscript reports the investigation of blends made of chitosan and polylactic acid, in which graphene oxide has been used as a compatibilizer agent to improve the miscibility between the components. The authors report a wide analysis of the chemical and physical properties of the materials, and a preliminary test on its antimicrobial effects, highlighting the role of graphene oxide in promoting superior performance.
In my opinion, the article is still not ready for being published in the Polymers journal because of the need to revise many points:
Major revisions:
- the manuscript needs an extensive English revision in terms of grammar and syntax,
- the introduction is a little bit messy. For example, the properties of chitosan are repeated several times, being redundant. Even the introduction related to the need for a "compatibilizer" agent for immiscible materials must be more emphasized. Please explain more in detail the meaning of different polarities. Some references about the use of compatibilizer for polymer blends that can be taken as examples are:
- Ock, Hyun Geun, et al. "Effect of organoclay as a compatibilizer in poly (lactic acid) and natural rubber blends." European Polymer Journal 76 (2016): 216-227.
- Mao, Nguyen Dang, et al. "Polyethylene glycol functionalized graphene oxide and its influences on properties of poly (lactic acid) biohybrid materials." Composites Part B: Engineering 161 (2019): 651-658.
- Vannozzi, Lorenzo, et al. "Novel ultrathin films based on a blend of PEG-b-PCL and PLLA and doped with ZnO nanoparticles." ACS applied materials & interfaces 12.19 (2020): 21398-21410.
- In addition, the introduction on graphene is poorly detailed. Recently, this is a widely investigated material used for its peculiar properties, like mechanical, electrical and even lubricant. Some references about the use of graphene oxide for biomedical applications, and related properties are:
- Talebian, Sepehr, et al. "Electrically conducting hydrogel graphene nanocomposite biofibers for biomedical applications." Frontiers in chemistry 8 (2020).
- Trucco, Diego, et al. "Graphene Oxide‐Doped Gellan Gum–PEGDA Bilayered Hydrogel Mimicking the Mechanical and Lubrication Properties of Articular Cartilage." Advanced Healthcare Materials (2021): 2001434.
- Please also analyze the use of graphene oxide as a compatibilizer in the state of the art.
- Is the tensile strength the only important parameter to extract from the tensile tests? PLease the authors deepen the mechanical characterization by analyzing the elastic modulus and the elongation/compression at break. This will further help to discuss the role of graphene oxide in the manuscript.
- The characterization of graphene oxide is necessary to support the discussion about the amino/hydroxyl/carboxyl groups present in the material, which confer the blend important features.
- Figure 2. It lacks scale bars. Furthermore, please the authors highlight the role of GO and chitosan, which are the most effective elements. It is extremely difficult to interpret the image without knowing the scale bar
- Table 1 lacks statistical analysis. In general, it should be added to improve the quality of the content.
- Figure 10. it is not clear without a scale bar. In addition, the quality of the image must be decisively improved. It is difficult to understand why the inhibition zone is only present in b) and c) sub-images.
Minor revisions:
- Line 70-71. Please the authors clarify the acronym PA and TP. Are these the same?
- Section 2.2.1. It is not totally clear to emphasize the need to remove chloroform before acetic acid, possessing a lower boiling point.
- I would suggest moving the FTIR spectra of PLA and CS in the Supplementary Materials, and unify Figure 1a and 1b to better highlight the differences among all materials with and without graphene oxide
- In Figures 4, 5 and 6 it is not clear the differences among all the tested samples. Please increase the readability of such images.
- Line 309-310. "The higher maximum degradation temperatures may be
attributed to the increase in molecular weight due to interaction between CS and PLA". Please the authors highlight the meaning of the increase in the molecular weight. - Line 333-334. Is the different Tg of chitosan related to its different molecular weights? please discuss it in the text.
- Figure 7. Please change the percentage values to absolute ones, to compare them with values present in the state of the art.
- Figure 8. Please add an explanation to an overall reduction of the water absorption in the composite materials.
Reviewer 3 Report
The manuscript needs major revision.
Comments:
Abstract:
Should revise the abstract to clearly mention the objectives, major findings, and final conclusion of this study.
90/10CS/PLA-GO and 70/30CS/PLA-GO exhibits: Check grammar
1. Introduction
polyester commonly made from lactic acid from a renewable resource such as : Modify
the comparison of organic fillers which is chitosan and chitin and inorganic fillers which is titanium powders:Re-frame
CS and PLA was not miscible: Check Grammar.
for one main reason different polarities.: Re-frame
PLA/CS/TP adding TP: Re-frame
However, there is no research work has reported in CS/PLA-GO composite.: There are already some papers published elsevier with the same components. Few References: Fabrication of chitosan/poly(lactic acid)/graphene oxide/TiO2 composite nanofibrous scaffolds for sustained delivery of doxorubicin and treatment of lung cancer;- https://doi.org/10.1016/j.ijbiomac.2017.08.048 & https://doi.org/10.1016/j.apsusc.2017.11.191
This method offers new: What method?
2. Materials and Methods
and acetic acid were used as analytical grade were supplied from Sigma-Aldrich. : Check grammar.
2.2. Synthesis of composites 2.2.1. Synthesis of CS/PLA films : Should be more specific is it composite or film? Both differ, makes different sense, so revise throughout MS.
90% w/w of chitosan (CS) was dispersed in 100 mL of acetic acid : 90% is a really high concentration, what was the reason to use such high concentration?
2%w/w of graphene oxide (GO) was added into PLA solution in chloroform: GO tends to dissolve in solvents with a high polarity such as water, PC, DMSO. Chloroform has a weak polarity, so GO has little solubility in chloroform.
Synthesis of CS/PLA-GO films: Synthesis is not an appropriate phrase, cant synthesis films, should fabricate them.
corresponding ethical approval code.: Details
Include section Statistical analysis
3. Results
While the peak at 1755 cm−1 159 is, and is markedly wider, there are obvious interactions among: Try to use past tense when describing the result
smooth surface composite indicate there is the interaction between CS and PLA : On what basis, you come to this conclusion? How smooth surface links to the interaction between molecules?
except for 50/50 composites. : Why? need clear explanation
Figure 2. Should provide uniform magnification for all groups. The images provided in Fig.2 were inconsistent. Images a,b,e,f,g and h were captured at 500 X (scale bar: 10 micrometers), whereas Images c and d were captured at 50 X with scale bar of 100 micrometers. Revise.
two characteristics peaks at 10.28°, 11.7° and 18.28°. : Here mentioned three data.
CS shows...10.28°, 11.7° and 18.28°: Check with Fig.3. The peaks appeared in different frequencies.
CS while the crystalline peak at 18.28° : But the axis showed up to 16° in Fig.3
crystal lattice (110, 040): Explain what was the number denotes for?
90/10CS/PLA showed the crystalline peak of CS and PLA: Mention peak details (data)
as was observed by [24][25].: What is this?
As for GO peak that was reported in Kalairasi et. al.,: This is really not standard scientific writing.
the broad peak was observed at 2θ ≈ 26° does not showed: Need to consult some native English scholar to refine the whole manuscript.
Diffraction peaks disappear in composites: When describing your results, always use the past tense.
reduces the CS loading has : Change
As can be seen in Figure 3, distinct broad peak at 8.84° : There is no peak at this wavenumber in Fig.3. Check
The whole para describing XRD data and results and justification is clumsy. Revise completely.
From figure 4: From Figure 4
The weight loss of the sample at 600 °C was 77 %: for which sample? The Fig. 4 shows peaks up to 500 °C, not 600 °C. Revise the Fig.4
where the maximum decomposition temperature is 325°C [27]. : Is this a present finding or an earlier report?
There is no residue & CS/PLA-GO composites is higher: Again same issue, grammar
left behind as all the materials decomposed by 535 °C: But previously you stated: " weight loss of the sample at 600 °C was 77 %."
composites increase the efficiency interaction between the two components. : Should give some supporting references.
with PLA prior before: Check
Figure 6 exhibit the DSC: Grammar
if there is no interaction existed ... will show two onset temperatures: Change
have only one T-onset thus deduce that there are number of interactions existed between CS and PLA : How it could be? Only one T-onset makes multiple interactions between CS and PLA?
Tonset (the onset temperature of thermal degradation) and Td (temperature at maximum degradation rate): Keep in the first usage.
Tg CS is said to be not sensitive : Change
As can be seen the Tg for 90/10CS/PLA is, The Tg for CS/PLA with GO composites is, compatibility and adhesion between CS, GO and PLA, in the SEM images will leads, Water absorption characteristics .. is presented, water absorption rises and fall, study reported by [33], might be due to the lower amount of CS used could lower, As reported by [34] GO, has been proven that has high, only the inhibition zone of..were, The highest Td and Tg achieved were,GO reinforced CS/PLA composites has : All the above sentences were poorly written, improper and not clear. Revise them.
Figure 8.: Provide statistical significance and P-value. Also in all other Figures.
water absorption of CS/PLA.. decrement compared to CS: Contradictory with your finding: Refer Line 294: the several functional groups in the GO such as carboxylic acid helps to increase the hydrophilic property of the PLA. Therefore increasing hydrophilic property should increase water absorption.
All CS/PLA composites were shown no antibacterial: Why? Previously you said"CS is known to naturally possess at least a mild antibacterial effect "
against Bacillus Subtilis B29: Here you used only one bacterial strain, you should use at least three to four bacteria in order to confirm the effect. Using one bacteria does not provide a convincing conclusion.
Figure 10. : Zones of inhibition were clearly seen in images. Especially in 90/10CS/PLA-GO and 70/30CS/PLA-GO composites which did not show inhibition zones.
Supplementary data: Fig.1s. FTIR of PLA sample.: This was already given in Figure 1a. So, delete this supplementary Fig.1
Figure 2s. FTIR of DSC sample.: What is this? FTIR of DSC?
Figure 3s. . FTIR of CS/PLA without GO samples: Check carefully, this is not FTIR data, should be DSC or TGA. Fig.3s already given in Figure 4. Delete.
Figure 4s. FTIR of CS/PLA with GO samples at different concentration.: Same, already given in text figure. so Delete.
Round 2
Reviewer 2 Report
The reviewer thinks that the authors only partially replied to the previous comments, leading very difficult to judge the increase in the manuscript quality and scientific soundness with respect to the previous version.
There is still the need of an extensive revision of English.
The suggestions provided to improve the introduction were not considered by the authors. For example, it is not clear why the role of "compatibilizer" of GO must be analyzed in another article, while the authors made a wide analysis on the role of GO within a CS/PLA blend.
For example, in Figure 2 the scale bar is totally missing being that on the image not visible to the reader. Or the quality of FIgure 10 is still the same as in the previous version.
Or in case of the "increase of the molecular weight", I think this is not justified by the text, because I can understand if a sort of cross-linking between PLA and CS happened, but it seems it was not demonstrated.
The stathistical analysis is totally missing.
I could understand the situation caused by the pandemic for more experimental tasks, but I do not see great efforts in such a revision task.
Reviewer 3 Report
Please consult with some professional people and native speakers to refine your manuscript. There are plenty of mistakes that still exist. The findings are well-conducted but very poor language misleads the overall merit of this manuscript.
I've corrected some sentences (only a few errors listed below, but you need to check completely throughout the paper)
42 spoilage, pathogenic bacteria, and fungi are: spoilage. The pathogenic bacteria and fungi are...
44 The chemical structure of CS comprises of one -NH2 group
and two –OH groups on every glucosidic residue [2] and the NH2 group in chitosan: Not clear.
CS-based composite is known as 'a materials': Wrong sentence structure.
materials that has: Use a plural verb
Chitin as second most abundant natural polymer after cellulose, and CS is derived from : Chitin is the second most abundant natural polymer after cellulose, and CS is derived from..
52, it is expected.: What expected?
61 due to CS hydrophilic properties pH sensitivity and the stability of the CS is: the CS is generally low due to hydrophilic properties and pH sensitivity.
67 renewable resources of lactic acid such as starch : What is the connection of lactic acid with starch?
67 It is a thermoplastic, high-strength: It means chitosan or PLA or starch or lactic acid?
73 in which it is non-polar and covalent bond: which is non-polar and covalent bond
83 adding compatibilizer or third polymer: adding compatibilizer or other polymer
86 PLA/CS/TP adding TP has significantly increased the heat sealing strength, water vapor permeability, and solubility of the composite membrane:Change to "addition of TP had significantly increased ...solubility of the composite PLA/CS/TP membrane.
87 there is no research work has reported: Please carefully check the grammar before submit your revised version.
89 CS-based other members: What you mean other members?
90 graphene oxide (GO) were : was. Again grammar issue. check carefully.
95 Graphene oxide also have: has
Herein, the aim of the study described here : delete 'here'
107 results will eventually open: results might eventually open
140 Spectrometer made in China: Spectrometer, City name, China
144 electron microscope (SEM) made in Japan:electron microscope (SEM), City name, Japan
4400 Universal Tester made in China: 4400 Universal Tester, City name, China
186 sliding calipers or a ruler was : sliding calipers or rulers were
301 The crystalline peak around 16.78° corresponding to a: Grammar missing. was corresponding..
311 there was has improvement: Really poor written.
324 As for GO peak that was reported in [27], the broad peak was observed at 26° does not showed in any XRD patterns.: Completely wrong sentence. Grammatically poor. Change to "the broad peak at 26° did not showed in any XRD patterns.
329 From figure 4: From Figure 4, can be seen that : it can be seen that
349 Meanwhile in Figure 5 shows: Meanwhile, the Figure 5 shows
351 the temperature at maximum degradation rate (Td) in CS/PLA-GO composites are higher: is higher
By increasing hydrophilic property of the PLA help... the interaction between the two components.: Wrong statement. write grammatically.
that there are interaction: Check grammar
376 there are three stage,: stages
378 Thus, as the molecular weight increase due to interaction between the CS, PLA and GO the higher maximum degradation temperature. : No grammar.
380 This is due to the molecular chains of the CS and PLA might
be deeply entangled : Check grammar
419 there are a few works concerning this Tg of CS gave quite different,: Check grammar
532 More details analysis of the biological performance of the
composites will be in our next step.: More studies are needed to investigate the biological performance of the composites, which will be done by our future research works.
Round 3
Reviewer 2 Report
I appreciate the time spent by the authors to comply with my comments, but sincerely I do not understand the introduction of this sentence in section 2.3:
"Statistical analysis was carried out through Microsoft Excel and standard deviation is graphically represented in the figure."
if there is no track of statistical analysis in all the manuscript.
The number of analyzed samples must be explicated.
Reviewer 3 Report
The revised version is satisfactory and acceptable for publication.
Author Response
Reviewer 3
April 22,2021
Ref (1005312)
Title: Evaluation on Structural Properties and Performances Of Graphene Oxide Incorporated into Chitosan/Poly-Lactic Acid Composites: CS/PLA versus. CS/PLA-GO
To
The Reviewer 3
Re: Response to Reviewer 3 Comments.
Dear Sir/Madam,
Many thanks for sending us reviewers’ valuable comment for our manuscript “Evaluation on Structural Properties and Performances of Graphene Oxide Incorporated Chitosan/Poly-Lactic Acid Composites: CS/PLA versus CS/PLA-GO”.
Reviewer’s comment:
The revised version is satisfactory and acceptable for publication.
Author’s Response:
My sincere gratitude to the reviewer on your consideration to sign the report. I really appreciate the time you took to write such detailed reviews for my journal.
Sincerely yours,
SITI NOOR KAMILAH
Faculty of Science,
Universiti Putra Malaysia,
43400 UPM Serdang,
Selangor.
This manuscript is a resubmission of an earlier submission. The following is a list of the peer review reports and author responses from that submission.
Round 1
Reviewer 1 Report
- A parametric analyses of chitosan with PLA and PLA-GO is presented with potential drug delivery applications.
- The paper is relatively well written and analysed within the proposed methodology.
- The paper explores the effects of chitosan on PLA film performance and investigates the thermal, mechanical and physical properties, and antimicrobial resistance.
- The paper does not sufficiently establish the innovation or need for this research, especially as graphene oxide is not stable in physiological solutions, which undermines the rationale for conducting this work.
- There is a distinct lack of a (PLA) control in DSC, tensile strength, moisture absorption, and antimicrobial tests.
- Elsewhere, complete characterisation is required for all samples, not just a subset
- The ethical considerations in the manuscript are obscure.
- Specific comments
- Figure 2 should show the features of interest
- Figures 4 to 8 should be more legible. Consider proper use of colour, text size, and contrast when developing the plots.
- Colours assigned to specific specimens (e.g., blue for 50/50 CS/PLA-GO) should be consistent throughout the plots.
- Crystallinity analysis should be conducted to correlate mechanical and thermal properties
- Absorption and desorption should follow guidelines in the testing standard and plotted accordingly. Any non standard plots should be justified.
- Antimicrobial activity should be comprehensively analysed and discussed.
Author Response
Dear reviewer,
Thank you so much for your careful reading and constructive comments. Please find attached file, detailed point-by-point response to all comments.

Reviewer 2 Report
The manuscript polymers-1005312 presents the preparation and characterization of chitosan/poly-lactic acid doped with graphene oxide (CS/PLA-GO) composites. I recommend the publication of this manuscript in Polymers journal, after major revisions.
1.Introduction
- p. 1, L. 38: “CS-based composite” is not a polymer! Could be a material! Please revise the sentence!
- p. 1, L. 41-42: Chitin is the second most abundant natural polymer after cellulose, and chitosan is a natural amino-polysaccharide derived from chitin! Please revise the sentence!
- p. 1, L. 42: “Generally.,”! Please revise this!
- Materials and Methods
2.1. Chemical and reagents
- Please also add the Mw of chitosan!
- p. 3, L. 97: Please revise: “the drying oven at 62 oC”!
- p. 3, L. 116: Please revise the sentence and avoid the repetition: “The structure of samples was examined using X-Ray diffraction using Shimadzu 6000 X-Ray machine”! XRD gives information on the crystalline structures of polymers! Shimadzu 600 X could be a diffractometer or an equipment!
- In my opinion the sections 2.3. Composite characterizations and 2.4. Composite performance analyses should be merged because both refer to the characterization of composites.
- Results
3.1. Structural properties of synthesized CS/PLA and CS/PLA-GO composites
FTIR
- The FTIR spectrum of chitosan is a bit confused, due to the fact that the broad peak between 3400 and 3700 cm−1, that corresponds to O-H stretching, it is barely visible! I consider that would be better if Figure 1 were improved! Moreover, it is a reversal between Figure 1a and 1b: (1a) PLA, and CS/PLA with GO, (1b) CS, and CS/PLA without GO.
- p. 4, L. 152-153: “the spectrum of CS clearly showed the asymmetric broadband due to the overlapping of O-H and N-H stretching bands extending from 3313 to 3278 cm−1”! This assumption isn’t sustained by the current Figure 1! The same observation for the paragraph L. 161-162.
- In Figure 1a is not clear where is the adsorption band at 1639 cm-1 from PLA, and even that one from 1536 cm-1 appear below 1500 cm-1. Please revise the figure!
- In Figure 1b I can’t identify where is the original band of the chitosan at 1643 cm-1! In my opinion and from what can I see in the current figure there is no band there!
XRD
- p. 6, L. 202: Please change the legend in Figure 3 and make the correspondence between the diffractograms and the lines in the legend!
- In my opinion, the XRD of the samples 50/50 CS/PLA and 70/30 CS/PLA seems the same, so please remove one of these.
- p. 6: The authors cannot discuss the XRD data of the composites compared to those of the initial samples (CS, PLA and GO) without presenting their diffractograms. Please add a figure with the diffractograms corresponding to the PLA and GO samples, or include them in Figure 3 (L. 212-213; L. 215-216)!
- p. 6, L. 216-218: How demonstrate the authors the increase of crystallinity of the samples? A reduction of the peak’s intensity doesn’t prove the increase of the crystallinity, but a decrease of it! The XRD section of this manuscript should be carefully revised!
TG/DTG
- It is very difficult to track the thermal degradation of composites in Figure 4. Please try to emphasize the influence of each added component, maybe even in different figures, to make it easier to identify the TG and DTG curves!
Degree of swelling
- Please explain the evolution of the swelling in water of the samples (Figure 7)! Why in the first three days the percent of water absorption increase, while for the last two days it was recorded a decrease? How the authors can explain this trend, if the samples were kept in water for five days and weighed at each 24 hours?
Author Response

(The authors gave the same response as above.)

Reviewer 3 Report
After the revision of the manuscript submitted to Polymers by Mohamad et al. entitled “Evaluation on Structural Properties & Performances of Graphene Oxide Incorporated into Chitosan/Poly-lactic acid composites: CS/PLA vs CS/PLA-GO
Introduction
In general, the introduction is based on the description of each of the polymers used in the work. But it does not highlight the novelty of the article, the properties of the final material to be achieved, it should establish more the state of the art more focused on the alternative 'drug delivery application' that is proposed.
Materials and methods
2.2.1. Synthesis of CS/PLA films
What final solid concentration was achieved before the solvent casting? Pay attention to the temperature units (°C).
2.3. Composite characterization
- “..XRD analysis was at the range of 0-40°C at the rate of 1°C/min”. The XRD analysis is not performed in temperature range but scanning angle. Correct the method explanation
Results
- Figure 1 must be deeply improved. The quality of the presentation format is not good enough. Why transmittance is about 190%? FTIR spectra are recorded from 100 – 0%. The units must be normalized if the spectra are displaced.
- Figure 3. If XRD intensity is showed in arbitrary units not values must be expressed.
- Figure 4. The figure needs to be improved. If the TGA analysis was performed to 600ºC, why not all the temperature range is showed? Numbers of DTG is hidden in the left graph. It must be deeply improved.
Author Response

(The authors gave the same response as above.)

Round 2
Reviewer 2 Report
In my opinion, even the authors generally paid attention to the comments, they didn’t make improvements to the manuscript polymers-909472, as compare to previous version:
- “Shimadzu 6000 X-Ray diffractometer” not “Shimadzu 6000 X-Ray machine”!
- XRD: “rate of 1 °C/min”??
- Perkin-Elmer, Pyris 7? I suppose that is TGA equipment! Moreover, Pyris is a software! It is necessary to be use the correct and complete name of the equipment and add information related to the company and the country for all equipment (as for example: Perkin-Elmer Corp., Norwalk, USA)!
- In addition, the authors don’t present anything about the DSC equipment! Please add all the adequate information!
- Even if I mentioned in the previous revision that the caption of Figure 1 (“FT-IR spectrums of (a) PLA, and CS/PLA without GO, (b) CS, CS/PLA with GO 199 composites”) doesn’t match with the figures: Figures 1a (PLA, and CS/PLA with GO) and Figure 1b (PLA, and CS/PLA without GO), the authors didn’t make the correction! Please revise the figure!
- Figure 4 doesn’t show TGA-DTG (l. 257), but only TGA profiles of samples!!! In addition, the authors discuss the DTG (l. 266) profiles without mentioning the Figure 5. Generally, is an ambiguous explanation, difficult to follow.
- “The onset temperature, Tonset of CS and PLA are different significantly, which are 230 °C and 315 °C, respectively”??? From Figure 4 and 5 there is not such a big difference between the Ton temperatures of the samples! How did the authors arrive at the mentioned observation?
- In Table 1 should be explained from where do you obtained the data, TGA or DSC! Add an additional row to separate the TGA data to the DSC data, in the idea of making it easier to understand the table. Moreover, it would be good to add a column with the data obtained for the sample residues (r, %)!
Supplementary data
- “Figure 2s. FTIR of DSC sample”?? There is no DSC sample within manuscript?
- “Figure 3s. FTIR of CS/PLA without GO samples at different concentration”?? “Figure 4s. FTIR of CS/PLA with GO samples at different concentration”??? Did the authors consider a TGA as FTIR?
I consider that this manuscript still has serious flaws and does not meet the requirements of a journal like Polymers, therefore I can't recommend this work for publication.
Reviewer 3 Report
The changes realized are not deep as proposed, several mistakes are mantained in the revised manuscript.